# Environmental Correlates of Health-Related Quality of Life among Women Living in Informal Settlements in Kenya

**DOI:** 10.3390/ijerph16203948

**Published:** 2019-10-17

**Authors:** Samantha C. Winter, Lena Moraa Obara, Francis Barchi

**Affiliations:** 1Columbia School of Social Work, Columbia University, New York, NY 10027, USA; 2Department of Sociology and Social Work, University of Nairobi, Nairobi 00100, Kenya; moraaobara@gmail.com; 3Edward J. Bloustein School of Planning & Public Policy, Rutgers, The State University of New Jersey, New Brunswick, NJ 08901, USA; Francis.Barchi@rutgers.edu

**Keywords:** women’s health, informal settlements, slums, Kenya, health-related quality of life, SF-36, water and sanitation, correlates of health

## Abstract

Informal settlements (slums)—defined as residential areas lacking durable housing; sufficient living and public spaces; access to basic infrastructure, water, sanitation, and other services; and secure tenancy—are presumed to be poor health environments. Research in Kenya suggests that residents of these settlements have the worst health outcomes of any population, yet there is a paucity of research focused on the health and well-being of these residents. Even less attention is given to the role played by environment in health in these settings. The present study addresses these gaps by examining potential environmental correlates, specifically access to water and sanitation, of health-related quality of life (HRQOL) among 552 women in Mathare slum in Nairobi, Kenya. A Kiswahili version of the 36-Item Short Form Health Survey (SF-36) measured HRQOL. Results suggested that access to a toilet at all times was associated with every subscale of the mental health and general well-being domains of the SF-36. Primary water source was also associated with women’s HRQOL. Despite increasing efforts to expand sanitation and water access in informal settlements, more attention should be given to whether the interventions being introduced, which likely affect women’s psychosocial health, are appropriate for all residents, including women.

## 1. Introduction

Informal settlements (also referred to as slums) are often presumed to be poor health environments. Indeed, a recent comparative analysis found that higher rates of mortality and morbidity were associated with slums than with rural and other urban environments [1]. Poor health has been attributed to poor quality housing and infrastructure, and a lack of access to water, sewage, garbage collection, health care, and other basic services [2]. Lack of educational and employment opportunities and absence of public sector and law enforcement can foster violence and unrest in settlements and expose residents to greater health risks by limiting their ability to easily and safely navigate their environments and manage their health and daily tasks [3]. In low-income countries, lack of access to safe water, sanitation, and hygiene (WASH) combines to account for about 842,000 deaths per year [4]. Poor sanitation is linked to a number of communicable diseases, notably cholera, diarrhea, dysentery, and typhoid [5]. Women with limited access to safe sanitation and water face particular challenges, including reproductive tract infections, constipation, hemorrhoids, and dehydration, as well as a number of psychosocial stressors [6,7,8,9,10]. Research carried out in informal settlements in Kenya suggests that residents of these settlements have the worst health outcomes of any population in the nation [3].

Other researchers have observed that self-perceived health-related quality of life (HRQOL) is higher among women living in informal settlements than among their peers living in more developed urban areas [11]. These researchers speculate that this rather surprising finding stems from two phenomena: ‘self-selection’, in that women who migrate into informal settlements tend to be younger, and possibly in possession of a more positive outlook about their health, and ‘geographic advantage’, in that informal settlements are often located in central neighborhoods in close proximity to health services. In addition to these two hypothesized correlates, a number of sociodemographic attributes have been linked to the HRQOL of women living in informal settlements; in Kenya, identified correlations have included age, education, employment, marital status, economic status, and disease state [12,13].

Despite recognition that inadequate sanitation and lack of access to water, both defining attributes of informal settlements, are leading risk factors for disease and causes of mortality [4], surprisingly little research has been done to date on the relationship between these environmental conditions and health-related quality of life in informal settlements. The present study aims to address this gap by examining potential environmental as well as sociodemographic correlates of HRQOL among women residing in Mathare Valley Informal Settlement in Nairobi, Kenya. Of particular interest is the extent to which sources of water and sanitation observed variability in HRQOL.

## 2. Materials and Methods

### 2.1. Research Setting

This study was carried out in 2018 in Mathare Valley Informal Settlement (Mathare), one of the oldest and largest slums in Kenya. The settlement started in the 1920s during the early years after the British established the Colony and Protectorate of Kenya. Colonial entities wanted to deter rural Kenyan nationals from relocating to urban centers by prohibiting the provision of housing and basic services to these migrants [14]. Thus, African residents in Nairobi were restricted to living in the flood-prone areas east and south of the city center [15]. In 1952 the British attempted to demolish the Mathare settlement. However, residents soon returned and there was an influx of new migrants. There are estimates that suggest that Mathare’s population was around 30,000 residents in 1969 and doubled by 1971 [15]. Today’s Mathare is comprised of 11 primary villages and is home to over 200,000 residents living in an area smaller than 3 square-kilometers [16].

Provision of formal water and sanitation infrastructure has been limited in Mathare. After a cholera outbreak in Mathare in 1971, the City Council started providing free water to some parts of the settlement, but it was and is still not adequate to service most residents [15]. Research from 2011 estimated that about 83% of residents in Mathare relied primarily on shared toilets used by approximately 85 households each. However, findings also suggested that over two-thirds of the participants in the study sample also periodically relied on plastic bags, buckets, or open defecation [17]. According to estimates from the same study, only about 11% of residents had access to private in-home or in-yard piped water while the majority of the remaining residents relied on stand points or water kiosks outside the home [15]. More recent estimates from research carried out with 550 female residents of Mathare in 2016 suggest that about three-quarters of female residents of Mathare rely on shared public toilets or ‘plot’ toilets shared by clusters of 20–30 households and about three-quarters of women use plastic bags, buckets, or open defecation to manage their daily sanitation needs at least once in a 24 h period [18]. Additionally, data from the same study indicate that almost 14% of female residents have access to private in-home or in-yard taps or wells, but over half (57%) of women rely on public taps (including kiosks) or wells for their drinking water [19].

### 2.2. Data and Sample

This study used cross-sectional data from a 2018 pilot project to establish a baseline understanding of women’s health in Mathare. The sample included 552 female residents in Mathare; approximately 50 from each of Mathare’s 11 villages. A well-documented technique for random sampling in informal settlements [17,20,21] was used with mapping software to identify households to be included in the survey. A single woman from each household was then selected using Kish [22] methodology. In order to be included in the study women had to speak either Swahili or English, be at least 18 years old, and be able to provide verbal consent. Surveys were administered by female residents in Mathare who have been working with the primary investigators on research projects in Mathare since 2015 and who have been trained extensively in data collection methodology and ethical research in informal settlements. The primary investigators were also on-site in Mathare throughout the data collection process.

### 2.3. Measures

Health-related quality of life (HRQOL) was measured using a Kiswahili version of the 36-Item Short Form Survey (SF-36) that had been previously translated and used in Tanzania (Wagner, et al., 1999). Although other Kiswahili versions of the SF-36 have been used effectively in Kenya [12,13], the Tanzanian version was selected for use in this study because its developers had conducted a psychometric evaluation of the items and empirically validated the scores as recommended by the International Quality of Life Assessment (IQOLA) [23,24].The SF-36 is a widely used questionnaire designed to capture self-reported data in eight major nonfatal physical and mental health domains [25]. These included three mental health subscales: social functioning (SF), i.e., limitations in social activities due to physical and or emotional problems; emotional role functioning (RE), i.e., limitations in one’s usual role because of emotional problems; and mental health (MH). As well as this, there were three physical health subscales: physical functioning (PF), i.e., limitations in physical activities due to health problems; physical role functioning (RP), i.e., limitations in usual role due to physical problems; and bodily pain (BP). Two further subscales to capture self-perceived general well-being included general health (GH) and vitality (VT), i.e., energy and fatigue. Subscales were scored using the RAND 36-Item Short form Survey scoring method [26]. Final subscale scores range from 0 to 100, with higher scores indicating better quality of life within the subscale.

#### 2.3.1. Water and Sanitation

Women were asked several questions about their primary and alternative sources for their water and daily sanitation needs. Answers to these questions were distilled into two variables for this analysis. First, answers to survey questions about women’s primary source for drinking water were collapsed into a 4-category ‘access to water’ variable (1 = relies on private tap/well inside home or yard, 2 = relies on shared tap or well outside home or yard, 3 = relies on public tap or well, and 4 = relies on water vendors, tankers, or burst pipes). Second, a binary sanitation variable was created by giving participants a score of ‘1′ if they stated having access to a toilet (of any kind) at all times throughout the day and night and a score of ‘0′ if they reported having to use a bag, bucket, or open defecation at least once in a 24 h period.

#### 2.3.2. Socioeconomic and Demographic Characteristics 

Finally, a number of socioeconomic and demographic variables including household income, education, marital status, number of children, age, household count, gender of head of household, employment, and business ownership were included in the models.

### 2.4. Analysis Strategy

Descriptive statistics were run for all variables of interest. Nonparametric Mann–Whitney U tests and Kruskal–Wallis tests were run to test for significant associations between each of the socioeconomic and demographic covariates and women’s HRQOL (see Appendix A). All socioeconomic and demographic variables were significantly associated with at least one of the eight subscales of the SF-36. A set of two linear regression models was run in Stata statistical software (version 15) [27] to look at the relationship between women’s access to sanitation and water and each of the eight subscales of the SF-36. Model 1 included only socioeconomic and demographic variables as independent variables and Model 2 included women’s access to sanitation and water while controlling for the socioeconomic and demographic covariates. Wald tests were used to compare model fit between Models 1 and 2 (see Appendix B). In order to account for the stratified nature of the sample across villages, the descriptive statistics and models were adjusted using the complex survey commands (svy) in Stata.

## 3. Results

### 3.1. Sample Characteristics

Sample characteristics are summarized in Table 1. Almost one-half of the women in the sample had monthly incomes between 5000 and 9999 KES (approximately US $50–100) and an additional 29% had incomes between 10,000–14,999 KES per month. Approximately 45% of the women had less than a complete primary education and only 5% of women had completed secondary school. Almost half of the sample was married and 35% reported being single or having a casual boyfriend. Almost 87% of women in the sample had at least one child. About three-quarters of the women lived in households with five or more people and almost half of the women lived in male-headed households. Women in the sample ranged from ages 18 to 75 years, with a mean age of 35 years. Over 58% of women were employed and about 34% of women ran a business. About 40% of women reported having access to some type of toilet at all times throughout the day and night. More than half of the sample reported relying on public taps or wells and an additional 34% reported using shared taps or wells outside of their homes or yards. Only 6% of women reported having access to a private tap in their home or yard.

### 3.2. Physical Health

A summary of the results from linear regressions of the physical heath subscales on women’s access to sanitation and water are presented in Table 2 with more detailed results, including nested model statistics, presented in Appendix B. After adjusting for socioeconomic covariates including income, education, marital status, number of children, age, household size, and employment, there were no significant associations between women’s primary sources for water and daily sanitation needs and women’s self-perceived physical health outcomes. 

### 3.3. Mental Health

A summary of the results from linear regressions of the mental health subscales on women’s access to sanitation and water are presented in Table 3 with more detailed results, including nested model statistics, presented in Appendix B. First, results from the Wald tests suggest that including women’s access to sanitation and water results in statistically significant improvement in the fit of the RE and MH models. Next, after controlling for key socioeconomic covariates, access to a toilet at all times was associated with a 6.7 point increase (CI = 0.22–13.18; *p* = 0.043) on the RE scale compared to women who relied on bags, buckets, or open defecation at least once throughout a 24 h period. Having access to a toilet at all times was also associated with a 4.7 point increase (CI = 2.05–7.41; *p* = 0.001) on the MH scale and a 4.0 point increase (CI = 0.49–7.55; *p* = 0.026) on the SF scale compared to women who relied on bags, buckets, or open defecation at least once throughout a 24 h period. 

Relying on a private tap or well outside one’s home but inside a plot was associated with a 6.7 point increase (CI = 1.24–12.09; *p* = 0.016) on the MH scale compared to women who relied on private taps in their homes or buildings. Relying on a public tap or well was associated with a 6.7 point increase (CI = 1.37–12.02; *p* = 0.014) on the MH scale compared to women who relied on private taps in their homes or buildings. 

### 3.4. General Well-Being

A summary of the results from linear regressions of the general well-being subscales on women’s access to sanitation and water are presented in Table 4 with more detailed results, including nested model statistics, presented in Appendix B. Results from the Wald tests suggest that including women’s access to sanitation and water in the regressions results in statistically significant improvement in the fit of the GH model. Additionally, having access to a toilet at all times was associated with a 3.0 point increase (CI = 0.02–5.99; *p* = 0.048) on the VT scale compared to women who relied on bags, buckets, or open defecation at least once throughout a 24 h period after controlling for key socioeconomic variables. Having access to a toilet at all times was also associated with a 4.8 point increase (CI = 1.68–8.01; *p* = 0.003) on the GH scale compared to women who relied on bags, buckets, or open defecation at least once throughout a 24 h period. 

Relying on a private tap or well outside one’s home, but inside a plot was associated with a 7.1 point increase (CI = 1.43–12.68; *p* = 0.014) on the VT scale compared to women who relied on private taps in their homes or buildings. Relying on a public tap or well was associated with a 7.3 point increase (CI = 1.99–12.65; *p* = 0.007) on the VT scale compared to women who relied on private taps in their homes or buildings. Relying on vendors, tankers, or burst pipes was associated with a 9.1 point decrease (CI = -17.79–(−0.39); *p* = 0.041) on the GH scale compared to women who relied on private taps in their homes or buildings.

## 4. Discussion

This study reports findings on one aspect of women’s psychosocial health, HRQOL, and its correlates in informal settlements in Nairobi, Kenya, paying particular attention to key environmental correlates. Evidence suggests that despite having better access to care and health service coverage in urban areas, residents of informal settlements have worse health on a number of indicators than members of other urban and rural communities [1,3]. Research also suggests that female residents of informal settlements over the age of 15 years may have a noticeable health disadvantage [28]. While there is a small, but growing body of research focused on health and the correlates of health in informal settlements, the studies have focused largely on child health and mortality, reproductive health, and maternal and prenatal conditions [3,28,29] with little attention given to psychosocial health and well-being. Even less attention has been given to correlates of health that take into consideration the role of environment on health, a critical gap in the literature given that lack of basic infrastructure and public services, particularly water, sanitation, and formal sewerage systems, are part of the very definition of informal settlements [30,31]. 

In their study focused on the role of characteristics of housing in health in informal settlements in Buenos Aires, authors French and Gardener observed that “human activity is not simply the product of abstract social norms and conventions, or an expression of inner, autonomous motives or drives: it is inextricably immersed in a physical, material environment that prompts, facilitates, constrains, and transmutes human action. The material world both configures, and is a product of, the everyday activity of humans” (p. 158, [32]). Findings from this study, exploring the role of access to water and sanitation in women’s HRQOL in informal settlements in Nairobi, highlight this critical link between a woman’s physical environment and her health. Findings suggest, for example, that while access to sanitation and/or water was not significantly associated with any of the three subscales of the SF-36 that make up the physical health domain, access to sanitation was significantly associated with every subscale of both the mental health and general well-being domains of the SF-36. Access to water was also significantly associated with the emotional well-being subscale of the mental health domain and both subscales that make up the general health domain of the SF-36.

Previous research in Mathare, carried out by the authors of this study, suggests that the majority of women living in these environments cannot, for a variety of reasons including lack of resources and/or time, fear of violence, health concerns, time of day, and building closures, access a toilet for all of their daily sanitation needs and, as a result, revert at least once in a 24 h period to using open defecation and/or plastic bags or buckets in their homes (which are then emptied into open drainages outside the home) to manage urination and/or defecation [18,33,34,35]. As more sanitation interventions are being introduced into settlements, women’s regular access to toilets is expanding. For example, findings from this study suggest about 40% of women report having access to a toilet at all times during the night and day, up from only 26% in 2016 [18]. Even the most recent evidence, however, suggests that well over half of women do not have access to a toilet at all times. This finding is important in itself, but even more critical is the evidence of the strong association between women’s access to a toilet at all times and their psychosocial health. Numerous studies have provided evidence that access to sanitation is linked to health [36,37,38,39,40], but this is the first study to explore associations between women’s access to toilets at all times and their HRQOL in informal settlements in Nairobi. The findings suggest that having access to a toilet at all times is significantly associated with better outcomes on the RE, MH, SF, VT, and GH subscales, all subscales falling under the umbrella categories of mental health and general well-being in the SF-36. These findings not only highlight the importance of readily accessible sanitation, but they also give us insight into a different dimension of women’s health in informal settlements that must be taken into account when designing interventions to improve health outcomes in this population. Specifically, if limited access to sanitation is a definitional attribute of informal settlements, then it must be a factor when considering the overall health and well-being of its residents, particularly women. 

Interestingly, neither access to sanitation nor water was significantly associated with any of the physical health subscales of the SF-36 in this study. This finding was surprising in two ways. First, given evidence suggesting there are strong associations between lack of access to sanitation and conditions like diarrhea, typhoid, and cholera [5], it seemed probable that lack of access to sanitation would be significantly associated with women’s reports about their physical health. However, our findings suggest there were no significant associations. Next, previous research has also shown that lack of access to an in-home or on-site water source is associated with poor health and pain for women [41,42]. Without in-home or on-site water women are frequently responsible for transporting heavy containers of water (e.g., jerricans or buckets) from the off-site source to their homes, leading to potential musculoskeletal disorders and pain [41,42]. In addition, poor water quality, which differs by source, has also been associated with negative physical health outcomes and conditions [43]. Findings from our study are inconsistent with these previous findings. Inconsistencies between findings in our study and others may be because most of the existent studies investigating links between access to sanitation and water and women’s physical health and pain were carried out in rural settings where the distance to toilets/sites for urination and defecation and water points are likely much greater than the distance most women travel to access these services in informal settlements. The inconsistency may also highlight limitations of the SF-36 measure. Questions in the SF-36, for example, are not likely to capture information about illnesses or conditions like those often associated with lacking access to sanitation or poor quality drinking water (e.g., diarrhea, cholera, and typhoid), particularly if these conditions were experienced more than a month in the past. The SF-36 only measures current and recent (in the past 4 weeks) HRQOL. This limitation of the SF-36 may also be problematic when exploring links between women’s access to water and their physical health. Geere et al. (2018), for example, suggest that women’s access to water may not be linked to current or recent experiences of pain (e.g., in the last week), but may be associated with chronic pain or injuries in particular parts of the body [42]. Using just the SF-36, we cannot test for associations between women’s access to water or sanitation and their experiences of chronic pain or previous injury or illnesses. More research, particularly comparative research between rural communities and informal settlement environments and using additional measures of physical health outcomes, should be done to explore potential associations between women’s access to water and sanitation and their experiences of chronic pain, injury, or other physical health outcomes. 

While physical health outcomes were not significantly associated with access to water, access to water was an important factor associated with other domains of women’s HRQOL. However, the nature of the associations differed by the type of water to which women had access. Access to a tap or well outside the home/yard, including public taps or wells, was associated with higher scores on the *MH* and *VT* subscales of the SF-36 compared to access to an in-home/yard water source. While in-home water taps/wells are often the gold-standard for water provision, there can be water quality challenges to in-home water taps/wells in informal settlements. Research suggests, for example, that relying on an in-home water tap or well may be associated with higher rates of diarrhea [19]. Literature suggests that providers of water at public kiosks and taps are usually licensed, and therefore, the water from those sites is typically regulated [44]. On the other hand, water from private vendors, individual households, and landlords who supply water to their tenants are usually unregulated; thus, there is little control over water quality [44]. In addition to water quality issues, there can also be water scarcity issues. Water supply in informal settlements is subject to rationing by the Nairobi Water Company. In informal settlements, water often flows from taps only three days out of seven and, when reservoir supplies are low, even fewer [45]. Water supply to individual households is often the first to be shut-off; thus, having access to more reliable private or public taps may have its advantages. In addition, water taps outside the home are often spaces for women to come together to socialize. Women congregate around water points to do laundry, fill jerry cans, and, while doing so, talk. Although there is a paucity of research focusing on the creation of social geography and important places within informal settlements, research does suggest that neighborhood connections in other environments is associated with positive psychosocial health outcomes [46]. 

Findings from this study also suggest that reliance on informal vendors/tankers/burst pipes is associated with lower scores on the GH subscale. During times of drought, formal water supplies are sometimes shutoff for a week or more at which point families have to buy water at an elevated price from water vendors selling water by the jerry can [45]. Other families may live too far from any tap/well or have a physical or health condition that limits their ability to fetch water, and, consequently, have to rely on vendors to deliver water to their homes. Water from vendors is expensive and often risky because buyers do not know the source or quality of the water. In light of these challenges, it is, perhaps, not surprising that reliance on informal vendors/tankers/burst pipes is associated with poor psychosocial health for women in these settlements.

While the findings from this study draw attention to an important and understudied aspect of health in informal settlements, the persistent environmental correlates of women’s HRQOL, the study has limitations. First, the data are cross-sectional; thus, we cannot make any causal claims about the relationship between the factors and women’s HRQOL. Additionally, the Swahili version of the SF-36 used in this study, while validated in a general Tanzanian population [47], has not been used widely in Kenya in general, and in informal settlements in particular; thus, the interpretation of some of the questions may differ somewhat in this population.

## 5. Conclusions

It is widely accepted that access to sanitation and clean and water and sanitation are critical to making improvements to health around the world. They were Target 7 of the Millennium Development Goals [48] and form Goal 6 of the Sustainable Development Goals [49]. Despite the acknowledgement of their importance to health, access to water and sanitation is persistently poor in informal settlements. Some research suggests that despite increasing efforts and interventions to expand coverage, the number of residents in informal settlements without access to clean and adequate water and sanitation may actually increase as the populations in these settlements continue to grow rapidly [50]. In sub-Saharan Africa, half of the urban dwellers already live in informal settlements, and this number is expected to triple by 2050 [51]. This is likely to have serious consequences for the health of residents in informal settlements, not just in terms of increased risk of communicable diseases such as diarrhea, cholera, and typhoid, but, as results from this study suggest, for women’s psychosocial well-being. The physical and social environment play a critical role in one’s health and well-being. Findings from this study highlight a need not just for greater access to sanitation and water in informal settlements, but for better and more appropriate access. Results suggest, for example, that women may not be able to take advantage of extant sanitation at all times during the day and night, which may have serious consequences for their psychosocial health. Additionally, results suggest that developers and policy makers need to pay attention to how the type of accessible water may also affect women’s psychosocial health. More research focused on the social geography in informal settlements is needed to examine how and where social spaces are created by residents and how these spaces influence residents’ overall psychosocial health and well-being—knowledge that could, in the future, be used to identify, develop, and target more holistic interventions to promote health and well-being in these environments.

## Figures and Tables

**Table 1 ijerph-16-03948-t001:** Sample characteristics (n = 552).

Socioeconomic Demographic Variables	Adjusted Proportions (%)
Monthly household income	
Less than 5000 KES/month	5.4
5000–9999 KES/month	47.1
10,000–14,999 KES/month	29.4
At least 15,000 KES/month	18.1
Aware of household finances	93.3
Education	
Less than primary	44.6
Completed primary school	23.9
At least some secondary	26.8
Completed secondary school	4.7
Marital status	
Married	46.2
Living with a man, not married	6.0
Regular partner, live apart	13.2
Single/casual boyfriend	34.6
Number of children	
No children	12.9
1–2 children	43.5
3–4 children	32.1
5+ children	11.6
Age mean (*standard deviation*)	35.0 yrs (*0.41*)
Household count	
1 person	10.7
2 people	14.3
3–4 people	38.6
5+ people	36.4
Head of household is male	47.6
Respondent is employed	58.3
Respondent has a business	34.1
Water, sanitation, and hygiene (WASH) variables	
Access to toilet	
Has access to a toilet at all times	40.0
Access to water	
inside tap/well	5.8
outside tap/well	34.1
public tap/well	50.7
vendor/tanker/burst pipe	9.4

**Table 2 ijerph-16-03948-t002:** Associations between water and sanitation and women’s physical health (n = 552).

Socioeconomic Demographic Variables	Physical Function (*PF*)	Physical Role Function (*RP*)	Bodily Pain (*BP*)
Monthly household income			
5000–9999 KES/month	6.62 ^†^	5.45	15.00 *
10,000–14,999 KES/month	6.23	3.67	15.06 *
At least 15,000 KES/month	4.05	10.65	17.10 **
Aware of household finances	9.28 *	8.04	4.29
Education			
Completed primary school	3.4 *	8.26 *	−0.67
At least some secondary	−0.30	3.20	−1.88
Completed secondary school	3.34	6.04	2.48
Marital status			
Living with a man, not married	0.68	−2.34	0.38
Regular partner, live apart	3.05	2.58	−2.50
Single/casual boyfriend	−1.19	−8.14	−4.79
Number of children			
1–2 children	−3.16	−2.68	−6.57
3–4 children	−3.02	−4.64	−7.50
5+ children	−2.24	−13.74	−14.55 *
Age	−0.61 ***	−0.67 **	−0.52 **
Household count			
2 people	2.96	−2.59	−3.42
3–4 people	6.44 ^†^	−1.58	−0.18
5+ people	8.01 ^†^	0.25	0.93
Head of household is male	1.93	1.41	1.03
Respondent is employed	2.05	8.46 *	4.11 ^†^
Respondent has a business	−4.01 *	−6.68 ^†^	−4.00
WASH variables			
Access to toilet			
Has access to a toilet at all times	1.58	2.35	2.80
Access to water			
outside tap/well	−0.07	0.72	−2.73
public tap/well	−0.26	−0.01	2.13
vendor/tanker/burst pipe	0.57	9.34	−2.26

^†^*p* < 0.1 * *p* < 0.05, ** *p* < 0.01, *** *p* < 0.001.

**Table 3 ijerph-16-03948-t003:** Associations between water and sanitation and women’s mental health (n = 552).

Socioeconomic Demographic Variables	Emotional Role Function (*RE*)	Mental Health (*MH*)	Social Functioning (*SF*)
Monthly household income			
5000–9999 KES/month	2.80	4.24	13.08 *
10,000–14,999 KES/month	1.13	7.89 **	12.32 *
At least 15,000 KES/month	3.67	7.90 *	12.06 *
Aware of household finances	3.64	3.00	2.73
Education			
Completed primary school	8.82 *	2.51 ^†^	3.01
At least some secondary	6.07	−1.65	2.38
Completed secondary school	−1.79	0.30	−0.42
Marital status			
Living with a man, not married	−6.85	3.74	0.94
Regular partner, live apart	5.20	−0.67	0.83
Single/casual boyfriend	−8.48	2.18	−3.00
Number of children			
1–2 children	0.96	−5.96 *	−9.62 **
3–4 children	−1.37	−8.80 **	−11.10 *
5+ children	−11.07	−15.93 ***	−21.53 ***
Age	−0.61 **	−0.01	−0.30 *
Household count			
2 people	−3.16	7.43 **	−0.54
3–4 people	−1.85	6.03 *	4.24
5+ people	0.88	7.68 *	8.87 *
Head of household is male	1.71	−0.86	−0.81
Respondent is employed	5.90 ^†^	2.90 ^†^	0.34
Respondent has a business	−8.29 *	−1.33	−4.33 *
WASH variables			
Access to toilet			
Has access to a toilet at all times	6.70*	4.73 **	4.02 *
Access to water			
outside tap/well	−1.30	6.67 *	7.26
public tap/well	6.06	6.69 *	8.55 ^†^
vendor/tanker/burst pipe	9.76	3.55	−1.37

^†^*p* < 0.1, * *p* < 0.05, ** *p* < 0.01, *** *p* < 0.001

**Table 4 ijerph-16-03948-t004:** Associations between water and sanitation and women’s general well-being (n = 552).

Socioeconomic Variables	Vitality (*VT*)	General Health (*GH*)
Monthly household income		
5000–9999 KES/month	−1.32	9.63 *
10,000–14,999 KES/month	0.13	12.47 **
At least 15,000 KES/month	−0.13	10.42 *
Aware of household finances	0.69	4.40
Education		
Completed primary school	3.42 ^†^	5.77 **
At least some secondary	−0.21	1.53
Completed secondary school	6.59	2.45
Marital status		
Living with a man, not married	2.84	−5.08
Regular partner, live apart	0.71	−2.66
Single/casual boyfriend	−2.37	−2.28
Number of children		
1–2 children	−1.49	−5.90 *
3–4 children	−10.79 **	−7.32 *
5+ children	−12.5 **	−12.83 **
Age	−0.11	−0.30 **
Household count		
2 people	6.16 ^†^	3.22
3–4 people	5.96 ^†^	5.09
5+ people	8.85 *	7.09 *
Head of household is male	−0.10	−4.29
Respondent is employed	5.14 **	0.04
Respondent has a business	−0.51	−1.43
WASH variables		
Access to toilet		
Has access to a toilet at all times	3.00 *	4.84 **
Access to water		
outside tap/well	7.06 *	−3.19
public tap/well	7.32 **	4.25
vendor/tanker/burst pipe	2.45	−9.09 *

^†^*p* < 0.1, * *p* < 0.05, ** *p* < 0.01, *** *p* < 0.001

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
