# Peer review of "Environmental Correlates of Health-Related Quality of Life among Women Living in Informal Settlements in Kenya"

_ijerph, 2019, doi:10.3390/ijerph16203948_

Round 1

Reviewer 1 Report

It would have been interesting for the reader to better see how sanitary and access to water compare to household and economic dimensions. The authors dont mention anything about these but age, number of infants and income do seem to play a very large role. A kind of multi-level regression or how much the Rsquare is improved by adding sanitary and access to water would have been interesting. I really like the use of SF36 but I would recommand the authors to better "label" the dimensions (at least in the columns to write the two letter labels in bracket.

The authors refer a lot to the literature showing the association btw sanitation and health but in their own study no such association btw perceived health and sanitation in significant (but with General health). The authors mention the non significant associoation but dont discuss it. I am a bit surprised. MAybe it could be explained by the fact that these "dimensions" reflect more the immediate health status and General health and mental health the chronic health but it needs to be addressed.

Author Response

Please find out detailed responses to the reviewers’ comments below. Thank you for the opportunity to respond to the comments and to make revisions to our manuscript.

It would have been interesting for the reader to better see how sanitary and access to water compare to household and economic dimensions. The authors don’t mention anything about these but age, number of infants and income do seem to play a very large role. A kind of multi-level regression or how much the Rsquare is improved by adding sanitary and access to water would have been interesting.

The reviewer raises an interesting point. Given the reviewer’s comments, we have revised the manuscript to include more detailed regression tables that report nested model statistics (results from Wald tests) and the r-squared values for each model. We would like to present these tables as an additional appendix (Appendix B) because they are rather complex tables to interpret but agree that they will be useful to those readers interested in a more detailed analysis.  We have updated the results section to include nested model statistics (see lines 173-177; 191-195). We also adjusted the analysis strategy to describe our method for comparing models (see lines 135-144).

I really like the use of SF36 but I would recommend the authors to better "label" the dimensions (at least in the columns to write the two letter labels in bracket.

In response to the reviewer’s comment, we have updated the tables to include the abbreviations for the SF-36 subscales, and have made sure that the abbreviations and naming of the scales is consistent throughout the text.

The authors refer a lot to the literature showing the association btw sanitation and health but in their own study no such association btw perceived health and sanitation is significant (but with General health). The authors mention the non-significant associations, but don’t discuss it. I am a bit surprised. Maybe it could be explained by the fact that these "dimensions" reflect more the immediate health status and General health and mental health the chronic health but it needs to be addressed.

We have added an additional section in the discussion specifically considering the non-significant association between women’s access to sanitation and water and the physical health domains of the SF-36. (see lines 259-287). 

Reviewer 2 Report

The research is presented in a very complete way. Originality, significance and quality are very clear. Moreover, the paper can be read pretty easily and the explanation is fluent.

In my opinion, the paper can be accepted with some really minor clarifications (or even in the present form). I would like to invite the authors to consider the possibility to:

1) add how the 36-Item Short Form Health Survey (SF-36) was applied (if an assistant helped the women to read and fill the survey or if they did it by themselves);

2) cite Table 2, Table 3 and Table 4 in text;

3) better explain the range 0-100 in the Subscale Scores (line 113) (0-100 seems to be the range of values that the women could express during the SF-36).

Author Response

Thank you for the comments and suggestions to help us improve our manuscript. We really appreciate the time you spent to review the document and to provide helpful feedback. Please find out detailed responses to the comments below. Thank you for the opportunity to respond to the comments and to make revisions to our manuscript.

Add how the 36-Item Short Form Health Survey (SF-36) was applied (if an assistant helped the women to read and fill the survey or if they did it by themselves)

This is an important oversight in our description of our methods and how data was collected. We have added a statement about how the data were collected (see lines 98-102).

Cite Table 2, Table 3 and Table 4 in text;

We have cited the tables in the results section (see lines 165, 174, and 192).

Better explain the range 0-100 in the Subscale Scores (line 113) (0-100 seems to be the range of values that the women could express during the SF-36).

We have provided a citation for the scoring of the subscales (see lines 117-118). Likert and binary responses are used as response options in the SF-36; however, these responses are combined, scored and transformed within their subscales such that the scores for each subscale range from 0 to 100. The reference we have cited details the scoring algorithms used to score the items on each subscale.